# Physical and Sedentary Activities in Association with Reproductive Outcomes among Couples Seeking Infertility Treatment: A Prospective Cohort Study

**DOI:** 10.3390/ijerph18052718

**Published:** 2021-03-08

**Authors:** Siret Läänelaid, Francisco B. Ortega, Theodora Kunovac Kallak, Lana Joelsson, Jonatan R. Ruiz, Julius Hreinsson, Kjell Wånggren, Anneli Stavreus-Evers, Ruth Kalda, Andres Salumets, Signe Altmäe

**Affiliations:** 1Institute of Family Medicine and Public Health, University of Tartu, Ravila 19, 50411 Tartu, Estonia; siret.laanelaid@ut.ee (S.L.); ruth.kalda@ut.ee (R.K.); 2Department of Nursing and Midwifery, Tartu Health Care College, Nooruse 5, 50411 Tartu, Estonia; 3Department of Physical Education and Sports, University of Granada, Carretera de Alfacar, S/N CP, 18071 Granada, Spain; ortegaf@ugr.es (F.B.O.); ruizj@ugr.es (J.R.R.); 4Department of Biosciences and Nutrition, Karolinska Institutet, NEO, SE-14183 Huddinge, Sweden; 5Department of Women’s and Children’s Health, Uppsala University, SE-751 85 Uppsala, Sweden; Theodora.Kunovac_Kallak@kbh.uu.se (T.K.K.); lana.salih@kbh.uu.se (L.J.); kjell.wanggren@telia.com (K.W.); anneli.stavreus-evers@kbh.uu.se (A.S.-E.); 6Minerva Fertility Clinic, Kålsängsgränd 10 d, 753 19 Uppsala, Sweden; julius.hreinsson@gmail.com; 7Department of Obstetrics and Gynaecology, Institute of Clinical Medicine, University of Tartu, L. Puusepa 8, 50406 Tartu, Estonia; andres.salumets@ccht.ee; 8Competence Centre on Health Technologies, Teaduspargi 13, 50411 Tartu, Estonia; 9Institute of Genomics, University of Tartu, Riia 23b, 51010 Tartu, Estonia; 10Department of Biochemistry and Molecular Biology, Faculty of Sciences, University of Granada, Avenida de la Fuente Nueva S/N, 18071 Granada, Spain; 11Instituto de Investigación Biosanitaria ibs.GRANADA, Parque Tecnológico de la Salud, 18016 Granada, Spain

**Keywords:** exercise, infertility, reproductive health, reproductive techniques, assisted, sedentary behavior

## Abstract

Background: The aim of this study was to investigate the association of physical activity (PA) with assisted reproductive technology (ART) treatment and pregnancy outcomes among couples seeking infertility treatment. Methods: This prospective cohort study was carried out among 128 infertile individuals (64 couples), entering the infertility clinic for ART procedures. Baseline PA (before entering any treatment) was assessed using accelerometry for both women and men. For every couple the infertility treatment outcomes were recorded. Results: The couples that required invasive ART procedures such as in vitro fertilization (IVF) or intracytoplasmic sperm injection (ICSI) spent less time in vigorous PA (−73 min/week per couple, woman + man) than those couples who became spontaneously pregnant after entering the study (*p* = 0.001). We observed no significant associations between the time spent in physical activities and positive pregnancy test or live birth. Conclusions: Our results do not support a positive nor negative relation between the time the couples spent in physical activities and the chances of getting pregnant or having a baby among patients seeking infertility treatment. However, couples undergoing invasive ART procedures did less vigorous PA than couples that became spontaneously pregnant, suggesting that PA may interfere with their reproductive health.

## 1. Introduction

Infertility is a failure to achieve pregnancy after regular unprotected intercourse for ≥12 months [1]. Around 48.5 million couples world-wide suffer from infertility [1,2] and it is steadily increasing with the trend to delay the time of first pregnancy in developed and developing countries [3], which might further increase the number of infertility cases in the future. Assisted reproductive technologies (ARTs) are widely applied for treating infertility, where in vitro fertilization (IVF) and intracytoplasmic sperm injection (ICSI) are the most effective and most used procedures.

The development of ARTs and the constant improvements in the treatment protocols have helped many infertile couples to achieve pregnancy, with over 8 million ART babies born today in the world and over 2.5 million cycles being performed annually, resulting in over 500,000 babies every year [4]. However, the success rates of pregnancy and live births among all ART-treated couples still remain ~30% per treatment cycle [5]. Identification of the novel ways to improve ART outcomes has become a critical topic for both clinicians and couples seeking infertility treatment.

Several factors can affect fertility, such as age and individual’s genetic background which are non-modifiable, while there is emerging evidence that different modifiable factors including physical activity (PA) and sedentary lifestyle can influence reproductive success and thus affect infertility treatment outcomes [6,7,8].

PA is a well-known therapeutic tool for various types of medical conditions [9,10,11] and WHO as well as other major health organizations recommend 150–300 min of moderate-intensity aerobic physical activity per week, in order to reduce the risk of different diseases [12,13,14]. In the reproductive field, the new WHO guidelines recommend pregnant women to engage in moderate-intensity aerobic physical activity for at least 150 min throughout the week [12]. However, couples attempting conception or couples seeking infertility treatment do not currently know whether higher/lower levels of PA can increase or reduce the chances of becoming pregnant and having live birth. In fact, no consensus has been reached regarding the effect of PA during ART on pregnancy outcomes [15]. The few studies of female and male PA in association with ART success have reported mixed results [15,16,17,18,19,20,21]. Nevertheless, the general belief is that doing regular PA is good for female [6,7,22] and male fertility [17,18,19,23,24], while high intensity and frequency of PA [25] or low PA or extensive sedentary time can increase subfertility and infertility [19,23,26,27,28]. Nevertheless, little evidence exists in this field at the moment, and major part of it comes from studies assessing PA using self-reported methods (mainly questionnaires) which are more prone to error and inaccuracies, which ultimately will underestimate the role of PA in the field of reproduction. As an example of this, the most recent and powerful study ever conducted on PA and mortality using accelerometers to objectively assess PA observed effect sizes two times larger than when studying the same association but using self-report methods [29]. These findings suggest that a new area of research using accelerometer-determined PA will shed light on the association between PA and health, and could be applied to the study of the role of PA and reproductive outcomes.

Further, previous studies on PA and fertility/infertility have analyzed either female or male partners of the couples separately [6,8,21,23,30,31], and no study has assessed the couples’ combined PA and sedentary activities on reproductive success. To our knowledge, the only study that has focused on the couples, analyzed different lifestyle factors including PA in couples seeking infertility treatment, while no joint effect of couples’ PA on ART was investigated [32]. Since succeeding in getting pregnant is a matter of both partners, studies investigating the combined levels of PA and sedentary time of both members of the couple are needed. The aim of the present study was to investigate the association of PA and sedentary time of the couple (the woman, the man and the combined couple’s level), objectively assessed by accelerometers, with the ART treatment and pregnancy outcomes.

## 2. Materials and Methods

### 2.1. Study Sample and Design 

Participants from the Physical Activity and Assisted Reproductive Technology (ActiART) project were enrolled in this prospective cohort study. In total, a convenience sample of 71 infertile couples agreed to participate, while valid data was obtained for 64 couples. The acceptance rate among this delicate group of patients was ~10%. The couples were recruited in the Centre for Reproduction at Uppsala University Hospital, Sweden between 2011 and 2014. The ActiART project was approved by the Regional Ethical Review Board in Uppsala (reference number: 2009/084) and written informed consent was obtained from each participant before entering the study.

The only inclusion criteria applied was to be a couple seeking for infertility treatment for the first time, meaning that the couple had been infertile at least for 1 year and had not undergone any previous infertility treatments. The exclusion criteria was to be single woman or homosexual couple. All participants filled in a questionnaire of general characteristics including educational level and smoking. Patient’s measurements of weight, height, and waist circumference were obtained by an assistant nurse. The BMI was calculated as weight (kg)/height2 (m). During the participants’ first visit to the clinic, an accelerometer was distributed together with a diary to document the wear- and non-wear times. Participants were asked to wear the accelerometer for the following 7 days, starting from the subsequent morning from the recruitment, and to remove it only while swimming, showering or sleeping.

Clinical records of infertility treatment(s) together with pregnancy data were obtained from the clinic between 2015 and 2018. Regular ART protocols of the clinic were followed as previously described [33,34]. Collected information from medical records included infertility treatment (primary infertility diagnosis, controlled ovarian stimulation, fertilization method, numbers of good quality embryos, and embryo transfer day), pregnancy (a serum pregnancy test was considered positive if β-hCG > 10 mlU/mL) and live birth. Cumulative pregnancy and live birth data were recorded.

### 2.2. Assessment of PA and Sedentary Activities

Uniaxial accelerometers GT1M (ActiGraph LLC, Pensacola, FL, USA) were placed on each participant at the hip level to objectively assess the duration and intensity of their physical and sedentary activities. We used the ActiLife software version 6.10.2 (ActiGraph LLC, Pensacola, FL, USA) to process accelerometer data. Data with zero counts over or equal to 60 min, with allowance for 1–2 min of counts between 0 and 100, were considered non-wear time [35], and therefore excluded. Days over 10 hours or more of registration time [35] were included in the analysis. Only couples wearing the accelerometer for at least 10 h/day for 3 or more days were included into the study (n = 65), since 3–4 days has been suggested to be informative of the individual’s habitual PA level [36]. One couple registered abnormally high levels of physical activity, i.e., 1554 min per week of moderate to vigorous physical activity (i.e., activity of the woman + men), which was 3.9 standard deviations higher than the average of the study sample. Therefore, even in the case of valid data, this couple was a clear an outlier in this study and was therefore excluded from the analyses.

The following physical and sedentary activity variables were derived from the accelerations registered as described in a recent meta-analysis [29]: total volume of PA (total counts/wear time in minutes, cpm); time (min/day) spent in six intensity specific variables: sedentary if ≤100 cpm, light if 101–1951 cpm, low light if 101–759 cpm, high light if 760–1951 cpm, moderate to vigorous if ≥1952 cpm and vigorous if ≥5725 cpm.

### 2.3. Statistical Analysis

Descriptive characteristics of the sample were expressed as means and standard deviations for continuous variables and as frequency and percentage for categorical variables. In order to test whether there was any difference on activity levels between couples who needed invasive treatments and who required no treatment (became naturally pregnant after entering the study), we categorized the different treatments into two groups, as described next. We used analysis of covariance (ANCOVA) to test differences on couples’ PA and sedentary time between the group receiving the invasive and prevalent treatments (i.e., IVF and ICSI) and the group that became pregnant spontaneously before being treated, after adjustment for couple age and accelerometer registration time. All couple variables (i.e., couple activity/sedentary variables, couple age and couple registration time) were computed as the sum of the individual values of the woman plus man within each couple. The associations of PA and sedentary time at the first visit to the clinic (before any treatment) with pregnancy and live birth (yes vs. no) were examined using binary logistic regression models after adjustment for age and registration time. Sensitive analyses were conducted including additional adjustment for other potential confounders, such as educational level. All statistical analyses were performed using the IBM-SPSS software, version 20.0, Armonk, NY, USA. The level of significance was set at *p* < 0.05 for all analyses.

## 3. Results

Characteristics of the study sample and baseline PA and sedentary activities among women and men before entering any ARTs in infertility clinic are presented in Table 1. In total 128 individuals, 64 couples, had complete physical and sedentary activity data for this prospective cohort study. Most of the participants met the current PA guidelines, i.e., 93% of the women and 98% of the men. Out of the 64 couples, roughly half-of the couples (N = 31) required invasive ART procedures (IVF or ICSI) and 27% (N = 17) became spontaneously pregnant after entering the study (Table 2). Considering the spontaneous conceptions and the cumulative ART treatments, in total 72% of women (N = 41) became pregnant, and 67% (N = 38) of the couples ended up with live birth.

When comparing the PA and sedentary activities between couples undergoing different treatment protocols, ANCOVA models showed that couples who needed invasive reproductive techniques (IVF or ICSI) spent less time in vigorous PA compared to the couples who became spontaneously pregnant and no treatment was required (*p* = 0.001) (Figure 1). Differences are shown in z-scores, which can be interpreted as number of standard deviations. For vigorous PA, the difference between groups was of 0.96 standard deviations, which is interpreted as a difference of large magnitude in accordance to the usual effect size interpretation (i.e., 0.2 small, 0.5 medium, 0.8 large). In addition, this difference is expressed in the original unit of the measurement, it was observed that the couples that became spontaneously pregnant accumulated a total of 73 min per week (95% confidence interval: 33 to 113 min/week) of vigorous PA more than the couples undergoing IVF/ICSI. No other variable of physical or sedentary activity differed significantly between the groups. In exploratory analyses, we conduct these analyses with additional adjustment for infertility diagnosis (grouped as female infertility, male infertility, unexplained infertility and missing information of diagnosis) and the results persisted identical (data not shown). Moreover, we computed a variable that express the proportion of the female activity to the summed couple activity to account for the contribution of the activity of each member of the couple to the total couple levels. As an example, the average of this newly created variable was 0.48 for vigorous physical activity, indicating that both members of the couple did, on average, a similar amount of activity at that intensity in our study sample. We conducted further testing on the difference in vigorous physical activity levels observed between couples undergoing IVF/ICSI and those who became spontaneously pregnant without any treatment. We observed that additional adjustment for this weighing variable did not alter the results, i.e., couples that needed IVF/ICSI did on average less physical activity of vigorous intensity than those that got spontaneous pregnancy (–73 vs. –76 min/week before and after this additional adjustment respectively, both *p* ≤ 0.001).

The associations of PA and sedentary time before any treatment with the chances of having a baby, after adjustment for age and accelerometer registered time, are shown in Table 3. No significant association was found between the individual (either woman or man) or couples’ PA or sedentary time and live birth. Additional adjustment for educational level of the participants did not alter the result and additional adjustment for smoking was not necessary since only 6 women and 3 men were smokers in this study sample. Likewise, when we used the positive pregnancy test (yes/no) as outcome, all the associations remained non-significant (data not shown). These two outcomes were nearly identical in our cohort, since all pregnancies ended up in live birth except for 3 cases which resulted in miscarriage (Table 2).

## 4. Discussion

Infertility treatment is costly in time, money and emotional stress, therefore it is of importance to identify factors which dictate the success of the infertility treatment. Especially appealing are the lifestyle factors that are modifiable such as physical and sedentary activities. To the best of our knowledge, this is the first study where couples’ physical and sedentary activities have been analyzed jointly in association with reproductive outcomes. Both partners contribute to the reproductive success or failure, and therefore it is important to understand the couples’ combined physical and sedentary activity levels on infertility treatment outcomes.

The main finding of our study is the no association (either positive or negative) of couples’ combined PA and sedentary time on infertility treatment outcomes, suggesting that physical and sedentary activities, at least within the levels of PA observed in the current study, might not influence the pregnancy establishment and outcome. Our findings do not support either an association of the separate physical and sedentary activity levels of the woman and man with the chance to get pregnant and to have a baby. This is in line with previous findings, where no associations of objectively measured physical and sedentary activities in women before and during IVF treatment with IVF outcomes such as implantation, clinical pregnancy and live birth were detected [21,37]. Additionally, a recent preconception prospective cohort study on 785 women with a previous history of a pregnancy loss analyzed self-reported PA and did not detect any effect of PA on the clinical pregnancy loss [38]. It could be that physical and sedentary activities are not harmful neither beneficial for the ART procedures to succeed, as is the case with natural conception, where women generally do not know when exactly they become pregnant and continue with their normal life-style, and for embryo implantation and pregnancy establishment PA at that exact moment would not be so crucial as is the general physical state/fitness of the woman. In fact, a number of studies demonstrate that bedrest, i.e., inactivity after embryo transfer in IVF programs does not influence pregnancy outcomes and is therefore unnecessary [39,40,41,42,43,44].

While no effect of physical and sedentary activities on reproductive success (positive pregnancy test and live birth) among couples seeking infertility treatment was observed, our study results demonstrate that couples that spent more time in vigorous PA required less invasive infertility treatment procedures. It is important to note that the magnitude of these differences was large, with more than 1 h per week difference in the time accumulated in vigorous PA between both groups. On a scale relative to an individual’s personal capacity, vigorous PA is usually defined as 7 or 8 on a scale of 0–10, in other words, vigorous PA included activities that make you breath markedly faster. For general audience, the “Talk Test” can be used to recognize when a person is doing vigorous PA. This test is based on the fact that you can still talk normally while doing PA of moderate intensity, whereas you will not be able to say more than a few words without taking a breath while doing PA of vigorous intensity. There is strong evidence supporting the benefits of vigorous PA for many health outcomes, and consequently, international PA guidelines recommend to include activities of vigorous intensity as part of weekly activity [13].

Although this is an observational study and causality need to be demonstrated in future randomized controlled trials, our findings support the notion that PA could positively interfere with couples’ general reproductive health. Additionally, previous studies are suggesting that PA may have beneficial effect on general reproductive health rather than on a specific ART treatment outcomes, like embryo implantation and pregnancy rate [21,37]. Evenson et al. detected beneficial effect of PA in women who were active in the year preceding infertility treatment having more likely favorable pregnancy outcome [37]. Sõritsa et al. study found that higher baseline PA and lower baseline sedentary activities were associated with better response for hormonal ovarian stimulation, while no effects on embryo implantation and pregnancy establishment were detected [21]. Similarly, PA has been positively associated with semen quality in men [18,20,45], but did not associate with higher reproductive success in infertility treatment outcomes [20]. These findings suggest that lifestyle habits such as PA might positively influence reproductive health in general rather than the process of pregnancy establishment. The effects of how exactly PA could influence human reproduction are not well studied, but different mechanisms are proposed: PA may influence hormone production [46]; PA regulates energy balance and affects BMI, which in turn, can influence reproduction [47]; PA can increase the expression of antioxidant enzymes throughout the body [48]; and PA can relieve stress and anxiety, which are shown to have negative impact on ART’s success [49]. Nevertheless, the knowledge of the thresholds for the amount and intensity of activities to achieve optimal fecundity is missing.

This study is subject to several limitations. First, the relatively small sample size analyzed, which may have reduced the power and generalizability of the findings. Additionally, the observational nature of this study does not allow to draw causal conclusions. Next, accelerometers used in the study required removal for water-related activities (so water-based activity cannot be recorded) and do not capture well activities such as cycling. To also note, new accelerometers are able to store raw accelerations at high frequency sampling, providing more detailed data and also the possibility of other data processing approaches [50], an option which was unavailable in our study. Nevertheless, the unique characteristic of the current study is that the couples’ physical and sedentary activities in this delicate group of patients provide the opportunity to evaluate the associations with both individual (women and men separately) as well as couple’s combined activity levels in relation to reproduction outcomes. In addition, and despite the limitations indicated above, the objectively measured PA using accelerometers is the major strength of our study, providing more accurate data of movements/activities than questionnaires.

## 5. Conclusions

This study results indicate that physical and sedentary activities do not seem to increase or decrease the chances of having a baby in couples seeking for infertility treatment. However, more active couples required less invasive ART procedures and could therefore rely more on natural conception.

## Figures and Tables

**Figure 1 ijerph-18-02718-f001:**
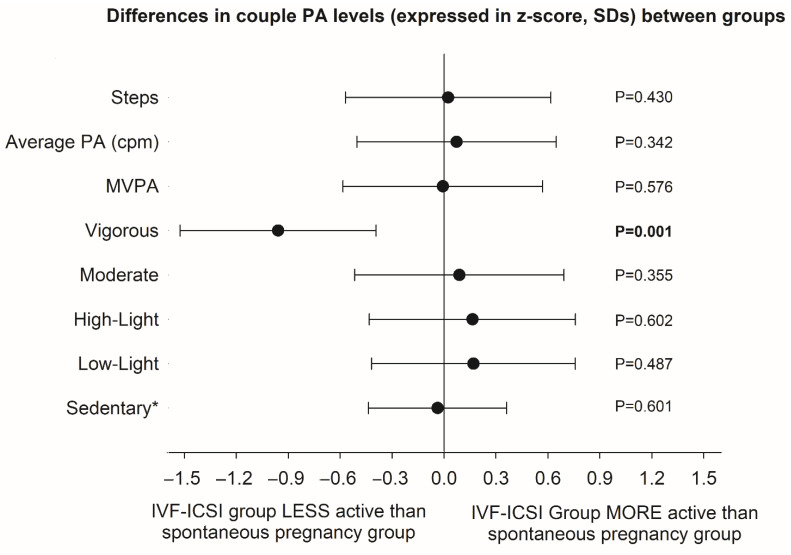
Physical and sedentary activity levels (expressed as z-scores, means and 95% confidence intervals) of the couples (accumulated time, i.e., sum) in those who needed invasive assisted reproductive techniques (IVF and ICSI) compared with those couples that became spontaneously pregnant. * Sedentary time score is interpreted opposite to the rest of the activity indicators. Differences between groups were analyzed by ANCOVA adjusting for couple age (sum of the woman and man ages) and accelerometer-registration time (sum of the woman’s and man’s registered time). PA–Physical activity, MVPA–Moderate to Vigorous PA, CPM–counts per minute.

**Table 1 ijerph-18-02718-t001:** Distribution of demographic and physical and sedentary activity characteristics of infertile couples.

Characteristics	Women		Men	
	N	(Mean ± SD) or %	N	(Mean ± SD) or %
Age (years)	64	32.4 ± 4.1	64	34.9 ± 5.6
BMI (kg/m^2^)	64	23.1 ± 3.7	64	25.4 ± 3.3
Waist circumference (cm)	64	75.9 ± 8.9	64	88.8 ± 10.2
Education N (%)				
Primary education	3	4.7	3	4.7
Secondary education	17	26.6	18	28.1
College education	8	12.5	11	17.2
University education	36	56.3	32	50
Occupation N (%)				
Working	54	84.4	56	87.5
Not working or studying	8	12.5	5	7.8
Missing	2	3.1	3	4.7
Smoking N (%)				
No	57	89.1	58	90.6
Yes	6	9.4	3	4.7
Missing	1	1.5	3	4.7
Snuff N (%)				
No	58	90.6	46	71.9
Yes	2	3.1	16	25.0
Missing	4	6.3	2	3.1
PA Characteristics (mean ± SD)				
Sedentary time (min/day)	64	683.5 ± 80.8	64	709.1 ± 65.0
Low-Light PA (min/day)	64	80.1 ± 24.7	64	77.2 ± 24.0
High-Light PA (min/day)	64	46.0 ± 16.9	64	44.9 ± 16.9
Moderate PA (min/day)	64	47.0 ± 16.8	64	48.0 ± 16.8
Vigorous PA (min/day)	64	5.0 ± 6.5	64	5.2 ± 6.6
Moderate-to-Vigorous PA (counts/min)	64	52.0 ± 19.1	64	53.2 ± 17.9
Meeting PA guidelines (%) *	64	92.9	64	98.2
Average PA (counts/min)	64	333.1 ± 117.5	64	316.9 ± 94.6
Steps (No/day)	64	7994.8 ± 2667.2	64	7456.3 ± 2093.9

BMI–body mass index, PA–physical activity, SD–standard deviation; * International PA guidelines recommend to do 150 min. per week of moderate-to-vigorous physical activity, which would be equivalent to 21.4 min. per day. The % of the study sample doing this amount of PA per day or more was computed.

**Table 2 ijerph-18-02718-t002:** Distribution of clinical factors for infertile couples and infertility treatment outcomes.

Clinical Characteristics	N	%
Primary infertility diagnosis		
Female factor	24	37.5
Tubal factor	9	14
Anovulation	6	9.3
PCOS	4	6.3
Endometriosis	3	4.7
POF	2	3.1
Male factor	11	17.2
Unexplained	22	34.4
Missing	7	11
Conception method		
IVF	22	34.4
ICSI	9	14.1
IUI	2	3.1
Ovarian stimulation	4	6.2
No treatment	5	7.8
Spontaneous pregnancy	17	26.6
Missing	5	7.8
Positive pregnancy test *	41	73.2
Miscarriage *	3	5.4
Live birth *	38	67.9

IVF–in vitro fertilization; ICSI–intracytoplasmic sperm injection; IUI–intrauterine sperm insemination; PCOS–polycystic ovary syndrome; POF–premature ovarian failure; * Data available for 56 couples.

**Table 3 ijerph-18-02718-t003:** Likelihood of having a baby according to time spent in sedentary time and physical activity in women, men and the accumulated (sum) activity level of the couples.

Z-Score Variables	Women			Men			Couples		
	OR	95% C.I.		OR	95% C.I.		OR	95% C.I.	
		Lower	Upper		Lower	Upper		Lower	Upper
Sedentary time	1.911	0.733	4.982	1.299	0.513	3.285	1.658	0.633	4.344
Low-Light PA	0.641	0.329	1.251	0.862	0.426	1.744	0.779	0.403	1.505
High-Light PA	0.788	0.416	1.493	0.790	0.385	1.621	0.954	0.513	1.773
Moderate PA	0.740	0.400	1.369	0.894	0.492	1.623	0.699	0.369	1.324
Vigorous PA	0.818	0.451	1.483	1.223	0.693	2.158	1.109	0.617	1.991
Moderate-to-vigorous PA	0.737	0.396	1.373	1.048	0.581	1.891	0.732	0.379	1.414
Average PA–counts/min	0.718	0.390	1.324	1.051	0.589	1.875	0.757	0.403	1.425
Steps	0.659	0.360	1.209	1.045	0.558	1.956	0.758	0.413	1.393

The variables were transformed into z-scores. Therefore, the interpretation of the estimates in the table reflect how much the OR increase or decrease by 1 standard deviation increment in the PA/sedentary variables. Logistic regression models were built using baby yes = 1 vs. no = 0 as dependent variable, and sedentary/activity variables as covariates. All models were adjusted for age and registered time. OR–Odds Ratio; CI–Confidence Intervals; PA–physical activity.

## Data Availability

The data presented in this study are available text and figure and tables.

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
