# Peer review of "Physical and Sedentary Activities in Association with Reproductive Outcomes among Couples Seeking Infertility Treatment: A Prospective Cohort Study"

_ijerph, 2021, doi:10.3390/ijerph18052718_

Round 1
Reviewer 1 Report
This study investigated the association of physical activity and sedentary time with fertility treatment and pregnancy outcomes. Sixty-four couples were included in the analysis. The main findings were that couples with more invasive fertility treatments (IVF, ICSI) had lower time spent engaged in vigorous activity and there were no associations between physical activity and positive pregnancy and live births.
This study provides novel information regarding fertility and physical activity for the couple as a unit. However, there are some minor concerns.
1- Table 3 outlines the infertility diagnosis however this information was not included in the analysis. Because of the physiological contributions of female factor vs. male factor including the infertility diagnosis as a covariate would strengthen the results.
2- page 3 line 148 describes combining the individual values of the female plus male for a composite couple score. With this method there is room for one partner to drive the activity. Please provide information showing there was no difference between male and females within a couple or account for possible weighting of one partner contributing more activity.
3- page 3 line 130-131. Please describe what "extreme values" mean.
4- Table 1 shows 7 female smokers and 6 males, page 5 line 196 says 4 females and 1 male smoker
Author Response
Comment 1
This study investigated the association of physical activity and sedentary time with fertility treatment and pregnancy outcomes. Sixty-four couples were included in the analysis. The main findings were that couples with more invasive fertility treatments (IVF, ICSI) had lower time spent engaged in vigorous activity and there were no associations between physical activity and positive pregnancy and live births.
This study provides novel information regarding fertility and physical activity for the couple as a unit. However, there are some minor concerns.
Authors’ response
We appreciate the positive comments.
Comment 2
Table 3 outlines the infertility diagnosis however this information was not included in the analysis. Because of the physiological contributions of female factor vs. male factor including the infertility diagnosis as a covariate would strengthen the results.
Authors’ response
We thank the reviewer for this constructive comment. We followed the suggestion. As shown in Table 2, the number of cases in the different female diagnosis is relatively low and to enter a nominal variable with 8 different codes (i.e. 7 dummy variables need to be entered into the model) reduce the degree of freedom markedly in the analyses. Therefore, we grouped the diagnosis variable into 4 groups: female infertility, male infertility, unexplained infertility and missing information on diagnosis. All the findings remained consistent after further adjustment for this diagnosis variable. Likewise, the significantly lower amount of time spent in vigorous physical activity observed in patients needing invasive treatments (i.e. IVF or ICSI) compared to those with spontaneous pregnancy, persisted after the additional adjustment for diagnosis type (i.e. -75min/week per couple, P=0.002). This information has been added into the Results’ section (see page 6, lines 195-198).
Comment 3
Page 3 line 148 describes combining the individual values of the female plus male for a composite couple score. With this method there is room for one partner to drive the activity. Please provide information showing there was no difference between male and females within a couple or account for possible weighting of one partner contributing more activity.
Authors’ response
Thanks for this interesting and constructive comment. Since the pattern of the associations between physical activity and likelihood of having live birth was consistently non-significant for both women and men separately, we believe that it is unlikely that weighing for one partner contribution to the couple activity levels would change the conclusions of this study. Nevertheless, in order to test this hypothesis, we computed a variable that express the proportion of the female activity to the summed couple activity, so that, for example, 0.5 would mean that both members of the couple contributed equally to the couple activity and 0.2 would mean that the woman contributed to the total activity of the couple with a 20%. We conducted further testing on the difference in vigorous physical activity levels observed between couples undergoing IVF/ICSI and those that became spontaneously pregnant without any treatment. We observed that additional adjustment for this weighing variable did not alter the results, couples that needed IVF/ICSI did on average less physical activity of vigorous intensity than those that became spontaneously pregnant (-73 vs. -76 min/week before and after this additional adjustment respectively, both P≤0.001). Interestingly, the average of this newly created variable was 0.48, indicating that both members of the couple did, on average, a similar amount of vigorous physical activity in the study sample. We have added this information to the manuscript, please see page 6, lines 198-209.
Comment 4
Page 3 line 130-131. Please describe what "extreme values" mean.
Authors’ response
The reviewer is right that further explanation on this was missing. There was one couple that registered abnormally high levels of physical activity, i.e. 1554 minutes per week of moderate to vigorous physical activity (activity of the woman + men), which was nearly 4 standard deviations higher than the average of the study sample. Therefore, even in the case of valid data, this couple was a clear outlier in this study sample and was therefore excluded from the analyses. This is now indicated in the Methods’ section (page 3, lines 132-136).
Comment 5
Table 1 shows 7 female smokers and 6 males, page 5 line 196 says 4 females and 1 male smoker
Authors’ response
Thanks for catching this typo. In order to clarify this error, we revised all database, and the correct numbers are 6 female smokers and 3 males, which is now consistently reported in the manuscript.
Reviewer 2 Report
It is difficult to assess the effects of physical activity on assisted reproductive outcomes. In a meta-analysis published in 2018, Rao concluded from 8 published studies that women's physical activity increased the rate of pregnancies and births, with no significant change in the rate of implantation and miscarriage. In addition, physical activity with cardiovascular exercise practiced more than 4 hours per week decreases the results of IVF (Morris 2006). In this study, on the contrary, it would seem that less intense physical activity is associated with more invasive procedures.
It would therefore seem that average physical activity is rather beneficial and the extremes rather negative. Can the authors further clarify these points of discussion?
Author Response
It is difficult to assess the effects of physical activity on assisted reproductive outcomes. In a meta-analysis published in 2018, Rao concluded from 8 published studies that women's physical activity increased the rate of pregnancies and births, with no significant change in the rate of implantation and miscarriage. In addition, physical activity with cardiovascular exercise practiced more than 4 hours per week decreases the results of IVF (Morris 2006). In this study, on the contrary, it would seem that less intense physical activity is associated with more invasive procedures.
It would therefore seem that average physical activity is rather beneficial and the extremes rather negative. Can the authors further clarify these points of discussion?
Authors’ response
Thank you for this interesting comment. We believe that the large differences in methodology does not allow direct comparisons and therefore the existence or not of discrepancies cannot be judged. The reviewer referred to the meta-analysis by Rao et al., 2018, in which the study included 8 studies. There was one intervention study testing the combined effect of exercise and diet, which in our opinion should have been excluded since the effect of exercise in isolation cannot be known, and the separate contribution of the exercise and diet interventions to the effect observed cannot be known. The 7 remaining studies were prospective, like ours, but all of them assessed physical activity by questionnaires instead of the objective assessment using accelerometers as was used in our study. Self-reported and objectively assessed physical activity largely differ and direct comparisons cannot be known. Anyhow, out of the 7 papers, 6 studied live birth and only 2 out of the 6 found a significant difference in live birth between active and inactive women. Therefore, the majority of the studies are in line with our findings supporting no association between physical activity before any treatment and future live birth. Also, the only previous study assessing the baseline physical activity using accelerometry before IVF treatment and pregnancy outcomes was conducted in Estonia by our group, and the results are consistent with the current study in Swedish population, no association between baseline physical activity and pregnancy rate or live birth (Sõritsa et al, 2020, doi:10.1007/s10815-020-01864-w).
The second study commented by the Reviewer, the one of Morris et al., 2006., is also based on self-reported data. In fact, the forest plot from the meta-analysis clearly show the study of Morris as the only one clearly deviated from the rest of studies. Anyhow, the main analysis shows no differences in live birth between women doing and not doing regular physical activity, also in line with our findings. Out of the different sub-group analyses conducted, the authors found a lower success rate in women doing more than 4 hours/week exercise during the last 1-9 years, but not during the last 10-30y. Being both indicators of regular and long-term physical activity, this lack of consistency in the findings is difficult to interpret. In our study, we only observed a difference particularly in the vigorous physical activity between those that needed IVF/ICSI treatment vs. spontaneous pregnancy, but Morris et al., did not conduct analogous analysis, so we can’t compare with their findings. In addition, Morris et al., did not accurately quantify the intensity of the activity, so comparisons with our significant findings cannot be done either.
Reviewer 3 Report
This study described the Physical and Sedentary Activities in Association with Reproductive Outcomes among Couples Seeking Infertility Treatment. It is interesting study, and the major concerns are described below:
- There is lack of information reading the study design. Is it a prospective study? Or the information was retrieved from a archive based on other study design?
- Inclusion and Exclusion criteria should be clear. Please, provide more detailed information.
- Have the authors performed a power analysis to assess the sample size?
Author Response
This study described the Physical and Sedentary Activities in Association with Reproductive Outcomes among Couples Seeking Infertility Treatment. It is interesting study, and the major concerns are described below:
Comment 1.
There is lack of information reading the study design. Is it a prospective study? Or the information was retrieved from an archive based on other study design?
Authors’ response
Thanks for the comment. Our study is a prospective cohort study, as stated in the beginning of the Methods’ section. The title indicated that it was a “longitudinal study”, which we have now changed to “a prospective cohort study” so that it is now clearer to the readers.
Comment 2.
Inclusion and Exclusion criteria should be clear. Please, provide more detailed information.
Authors’ response
Thank you. We have indicated more detailed information (please see page 3, lines 105-108): ‘The only inclusion criteria applied was to be a couple seeking for infertility treatment for the first time, meaning that the couple had been infertile at least for 1 year and had not undergone any previous infertility treatments. The exclusion criteria was to be single woman or homosexual couple.’
Comment 3.
Have the authors performed a power analysis to assess the sample size?
Authors’ response
No, it was a convenience sample and no a priori power analysis was conducted. However, looking at the effect sizes observed, there was a robust ‘no association’ indicating no association between baseline physical activity levels and pregnancy rate or live birth and even larger sample size with these effect sizes observed would have resulted in non-significant association. We have indicated in the Methods section that this was a convenience sample (page 3, lines 98-99).
Reviewer 4 Report
Thank you for the opportunity to review your manuscript titled Physical and sedentary activities in association with reproductive outcomes among couples seeking infertility treatment: a longitudinal study. I have some comments for the authors:
Keywords: Please use MeSH terms. Only 'infertility' is a MeSH term.
Materials and Methods:
Study sample and design. From the description of the sample selection, I deduce that it was a convenience sample. If so, please state it in the description. How many couples were invited to participate / possible in relation to those who participated? It would be interesting if they made a percentage of possible couples / couples that accepted.
Author Response
Thank you for the opportunity to review your manuscript titled Physical and sedentary activities in association with reproductive outcomes among couples seeking infertility treatment: a longitudinal study. I have some comments for the authors:
Comment 1.
Keywords: Please use MeSH terms. Only 'infertility' is a MeSH term.
Authors’ response
We thank the reviewer for the very good comment. The keywords have been modified and we used the following MeSH terms in our revised version: exercise; infertility; reproductive health; reproductive techniques, assisted; sedentary behavior.
Comment 2.
Materials and Methods: Study sample and design. From the description of the sample selection, I deduce that it was a convenience sample. If so, please state it in the description. How many couples were invited to participate / possible in relation to those who participated? It would be interesting if they made a percentage of possible couples / couples that accepted.
Authors’ response
The reviewer is right; it is a convenience sample. We have added this information into the Methods’ section (page 3, lines 98-100). The Centre for Reproduction at Uppsala University Hospital performs annually in total ~1000 fresh and frozen embryo transfer cycles, where the estimate of new couples is around 200-300 each year (including single women and homosexual couples). All the new couples entering infertility clinic were invited to participate, while only ~10% of the couples agreed to participate. These couples were often nervous, without clear diagnosis for their infertility problems, and were rather reluctant to participate in a research project.